

# Risk factors of self-reported physical child abuse during the COVID-19 pandemic in Japan: work-related changes in men and fear of COVID-19 in women

Haruaki Naito[1,2,*], Katsuya Nitta[1,*] and Yasuhiro Kakiuchi[1,2]

[1] Department of Forensic Medicine, School of Medicine, Kindai University, Sayama, Osaka, Japan
[2] Department of Forensic Medicine, School of Medicine, Tokai University, Isehara, Kanagawa, Japan
[*] These authors contributed equally to this work.

Corresponding author
Yasuhiro Kakiuchi,
kakiuchi@yokohama-cu.ac.jp

## ABSTRACT

**Background**. There is no global consensus on whether pandemic-related factors are related to child abuse. How the pandemic reinforces the risk factors of child abuse might depend largely on individuals' current and past lifestyles in each country. Some changes of lifestyles continue after the pandemic, and it is important to understand which factors are strongly associated with child abuse. We analyzed the pandemic-related characteristics of offenders and non-offenders of self-reported child physical abuse from Internet survey data in Japan and discussed how the pandemic affected physical child abuse and what caused the difference by gender.

**Methods**. We conducted a cross-sectional study on physical child abuse by caregivers based on the Internet survey conducted from September to October 2021. We divided the participants who were living with their child aged less than 14 years into offenders and non-offenders based on the answer to the question about physical child abuse. The population distribution of the sample was compared to that of caregivers under the same conditions in a large Japanese dataset. The association between their characteristics and physical child abuse was analyzed by univariable and multivariable analysis.

**Results**. The caregivers analyzed in the cohort had similar population distributions to those in the large Japanese dataset. As risk factors of male offenders, "work from home 4–7 days/week," "decreased work," "normal relationships with household members (compared to good relationships)," "COVID-19 infected, both themselves and household members, within a year," "unwillingness to receive COVID-19 vaccination because the license process of the vaccine is doubtful," "high levels of benevolent sexism," and "history of child abuse" were observed. As risk factors of female offenders, "bad relationships with household members (compared to good relationships)," "fear of COVID-19," "COVID-19 infected, both themselves and household members, within a year," "feelings of discrimination related to COVID-19 in the past two months," and "history of child verbal abuse" were observed.

**Conclusions**. Among male offenders, a significant relationship was observed regarding work-related changes, which may have been reinforced by the pandemic. Furthermore, the extent of the influence and fear of losing jobs caused by these changes may have varied according to the strength of gender roles and financial support in each country. Among female offenders, a significant relationship was observed regarding fear of infection itself, which is consistent with the findings of other studies. In terms of factors

related to dissatisfaction with families, in some countries with prominent stereotyped gender roles, men are thought to experience difficulties adapting to work-related changes induced by crises, while women are thought to experience intense fear of the infection itself.

## BACKGROUND

The negative impacts of child abuse are maintained throughout the victims' lifespan, and result in various mood disorders, anxiety disorders, suicidal behavior, and the intergenerational cycle of child abuse (*Chapman et al., 2004*; *Fujiwara & Kawakami, 2011*; *Stickley et al., 2020*; *Horikawa et al., 2016*; *Berlin, Appleyard & Dodge, 2011*). Many studies have concluded that more attention to child abuse is necessary during the COVID-19 pandemic because some risk factors such as mental stress, economic stress, social isolation, and a decrease in the chances of detecting child abuse must be reinforced by the pandemic (*Romanou & Belton, 2020*; *Pereda & Díaz-Faes, 2020*; *Brown et al., 2020*; *Amin & Parveen, 2022*).

However, there is a fundamental question of whether the pandemic equally reinforces the risk factors of child abuse in every country. A systematic review that analyzed 12 articles reported that the number of areas where abuse increased and decreased from pre-COVID-19 pandemic was approximately the same (*Rapp et al., 2021*). The impact of the COVID-19 pandemic on child abuse might largely depend on each country's past and current conditions, including the extent of fear of COVID-19, the degree of telework penetration, and attitudes toward gender roles.

Hence, we used Internet survey data and performed statistical analysis by including various factors directly and indirectly related to the COVID-19 pandemic. There are several drawbacks to using official reports or self-reports of child abuse concerning the quality and number of child abuse cases (*Gilbert et al., 2009*). However, it is plausible to estimate that self-reports were closer to the true number of child abuse cases during the pandemic when many child protection agencies and schools—major resources of official reports of child abuse—reduced their activities. This study is the first in Japan to report that some pandemic-related characteristics in offenders were significantly associated with physical child abuse; notable differences by gender were also observed. To the best of our knowledge, no study in Japan has identified a significant relationship between physical child abuse and pandemic-related factors of offenders.

## METHODS

### Data collection

Data was obtained from the Japan COVID-19 and Society Internet Survey (JACSIS) 2021. The target population of JACSIS 2021 consisted of the respondents of JACSIS 2020, the

respondents of New Tobacco Internet Survey (JASTIS) 2015–2020, and new participants. JACSIS survey focused on public health issues regarding COVID-19 pandemic and was conducted by an Internet research company (Rakuten Insight, Inc.), which had 2.3 million registered panelists. JASTIS survey focused on issues about new types of cigarettes in Japan and was also conducted by the same company from 2015. Before these surveys, the target number of responses was set in each group, stratified by age, sex, and prefecture, based on the population distribution. The population distribution was calculated using the Comprehensive Survey of Living Conditions (CSLC) in Japan. From September 27 to October 5, 2021, questionnaires with pre-defined options were distributed through the Internet to the 33,081 respondents of JACSIS 2020 and JASTIS 2015–2020 aged 16–81 years, in seven waves. From October 23 to 28, the target numbers were not reached in some stratified groups; therefore, questionnaires were distributed to the new 26,138 panelists by random sampling, based on the difference between the target and collected numbers in each group. In the second wave, all groups reached the target numbers and the survey was completed. Hence, questionnaires were distributed to 59,219 panelists, and the respondents represented the Japanese population regarding age, gender, and residential prefecture using a simple random sampling process (*The Japan COVID-19 and Society Internet Survey, 2022*).

Invalid responses were measured using three questions, Q18, Q26, and Q68, which were similar to the exclusion criteria used in previous JACSIS studies (*Yamada et al., 2021*; *Minoura et al., 2021*). In Q18, "Please choose the second option from the bottom," people who did not choose the correct option were excluded. In Q26 about 20 types of diseases, people who answered "currently" to all nine types of diseases (hypertension, diabetes, asthma, atopic dermatitis, allergic rhinitis, myocardial infarction or angina, stroke, cancer, and chronic pain over three months) were excluded. In Q68, regarding the use of alcohol and drugs, people who answered "almost every day" or "sometimes" to all the options *i.e.,* alcohol, sleeping pills, anxiolytic, illegal opioids, organic solvent, dangerous drugs, cannabis, stimulant drugs, cocaine, and heroin, were also excluded. From the 31,000 responses, 2,825 invalid responses were excluded, and finally 4,393 were analyzed according to the aim of this study. The detailed participants selection has been described in Fig. 1.

## Outcome variables
We focused on the question about child physical abuse:
 "Have you used violence on your child in the past two months?"
The question was only asked to caregivers living with child aged less than 14 years. Those who answered "Yes" were classified as an experimental group, and those who answered "No" were classified as a control group. Those who answered "not sure" or "not want to answer" were excluded from the analysis.

## Explanatory variables
The following sociodemographic characteristics of the subjects were obtained: "age (20–29, 30–39, 40–49, and 50–59 years)," "work from home (none, 1–3 days/month, 1–3 days/week, 4–7 days/week, and unemployed)," "change of job status within a year (did not change,

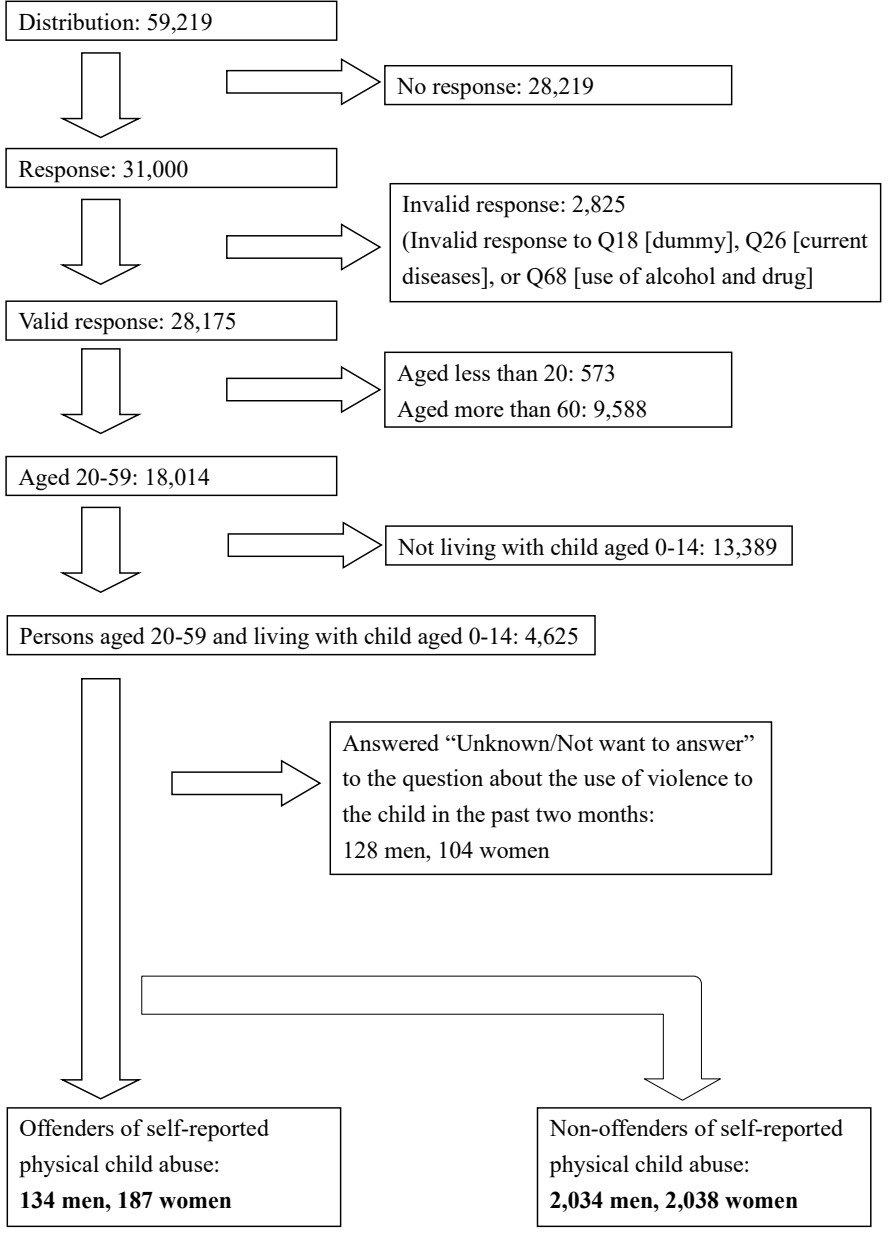

**Figure 1  Flowchart of the analyzed participants selection.**

work decreased, leave of absence or furlough, resigned, and others),'' ''relationships with household members (good, normal, and bad),'' ''benevolent sexism scale (<= 24 and >24),'' ''experienced or witnessed physical abuse by parents before age of 18 (no and yes),'' and ''experienced verbal abuse by parents before the age of 18 (no and yes).''

The following pandemic-related characteristics of the subjects were also obtained: ''fear of COVID-19 (not strong and strong),'' ''infection with COVID-19 within a year (neither myself nor household member, only myself, only household member, and both myself

and household member),'' ''feelings of discrimination related to COVID-19 (never, before two months ago, and in the past two months),'' and ''willingness to receive COVID-19 vaccination (yes, no because the license process of the vaccine is doubtful, and no due to other reasons).''

Regarding "fear of COVID-19," those who answered "very applicable" to any one of the questions regarding physically or mentally serious reactions to simply considering COVID-19 were defined as having a strong fear of COVID-19. Among the all reasons of "unwillingness to receive COVID-19 vaccination," "the license process of the vaccine is doubtful" is the most skeptical reason. Other reasons indicate that people did not want to receive the vaccination because they were concerned with the side effects, doubted its effectiveness, had low risk of serious complications, were previously infected, or did not have time to get the vaccine. "Benevolent sexism scale" was set to see whether the attitudes toward gender role are related to child abuse. Glick et al. proposed that sexism is divided into hostile and benevolent one in modern times, and that benevolent individuals have a more negative impact on women's career advancement than hostile ones (*Glick & Fiske, 1996*; *Dardenne, Dumont & Bollier, 2007*; *King et al., 2012*; *Jones et al., 2014*). The original sexism scale by Glick and Fiske consists of 11 questions each for hostile and benevolent individuals. In 2018, to make the scale more practical to measure benevolent sexism in Japan, a new benevolent sexism scale with eight questions corresponding to question numbers 1, 4, 8, 14, 15, 16, 18, and 19 in the original scale, was developed (*Morinaga et al., 2018*). The JACSIS 2019 fully used the scale in Q38. Victims of dating violence or child abuse had reportedly higher scores for benevolent sexism(*Cuadrado-Gordillo, Fernández-Antelo & Martín-Mora Parra, 2020*; *Vives-Cases et al., 2021*; *Melissa Abi, Ahmed & Rashid, 2021*). To our knowledge, common criteria for the benevolent sexism scale do not exist; we set 24 out of 48 as a cut-off value for the explanatory variable.

## Statistical analysis

First, our sample was compared to that of caregivers in the CSLC data in 2019 under the same conditions, to show the representativeness of the valid respondents. Second, Pearson's chi-square ($\chi^2$) test was performed to determine the significance of the overall variables. Third, a multivariable logistic regression analysis was performed to estimate the strength of associated risk factors to physical child abuse. Variance inflation factor (VIF) values were calculated to indicate that there was no multicollinearity issue among the variables. At 5% significance level, these gender-stratified analyses were conducted by EZR ver. 1.54.

## Ethics approval

Ethical approval was obtained from the Research Ethics Committee of the Osaka International Cancer Institute (approval no. 20084). Before answering the distributed questionnaire, in the first page of the questionnaire, all the participants read the purpose of the study, how personal information would be handled, and how to decline participation in the study. Those who answered the questionnaire were considered to have consented to participate in the study.

**Table 1** Epidemiological characteristics of caregivers from the Japan COVID-19 and Society Internet Survey (JACSIS) 2021 and the Comprehensive Survey of Living Conditions (CSLC) 2019.

|  | n (%) | Analyzed caregivers | Non-analyzed caregivers | Caregivers from CSLC |
|---|---|---|---|---|
| **Men** | **Number/Total** | 2168/13870 (15.6) | 128/13870 (0.9) | 35819/206437 (17.4) |
|  | **Age** |  |  |  |
|  | 20–29 | 105 (4.8) | 13 (10.0) | 1807 (5.0) |
|  | 30–39 | 820 (37.8) | 55 (43.0) | 12887 (36.0) |
|  | 40–49 | 1004 (46.3) | 53 (41.4) | 17033 (47.6) |
|  | 50–59 | 239 (11.0) | 7 (5.5) | 4092 (11.4) |
|  | **ISCED >=6** | 1503 (69.3) | 86 (67.2) | 13129 (36.7) |
| **Women** | **Number/Total** | 2225/14305 (15.6) | 104/14305 (0.7) | 40464/217960 (18.6) |
|  | **Age** |  |  |  |
|  | 20–29 | 222 (10.0) | 16 (15.4) | 2949 (7.3) |
|  | 30–39 | 952 (42.8) | 53 (51.0) | 16821 (41.6) |
|  | 40–49 | 930 (41.8) | 30 (28.8) | 18083 (44.7) |
|  | 50–59 | 121 (5.4) | 5 (4.8) | 2611 (6.5) |
|  | **ISCED >=6** | 969 (43.6) | 46 (44.2) | 8549 (21.1) |

Notes.
ISCED, International Standard Classification of Education.

# RESULTS

Figure 1 shows the flowchart of participant selection. Of the 2,296 male caregivers with children aged less than 14 years, 134 answered "Yes," 2,034 answered "No," and 128 answered "Unknown/Do not want to answer" to the question of physical child abuse over the past 2 months. Of the 2,329 female caregivers with children aged less than 14 years, 187 answered "Yes," 2,038 answered "No," and 104 answered "Unknown/Do not want to answer" to the same question.

Table 1 shows the comparison between our sample and that of caregivers from the CSLC data under the same conditions, aged 20–59 years with children aged less than 14 years. In our study cohort, of both male and female caregivers, 15.6% answered "YES" or "No" to the question. Of both male and female caregivers, 0.7–0.9% answered "Unknown" or "Do not want to answer." In the CSLC data, the caregivers under the same conditions were 17.1–18.6% men aged 16–81 years. Regarding age, the study cohort caregivers had a similar distribution to that of the caregivers from CSLC, compared with the non-analyzed caregivers. Regardless of the responses, the respondents of JACSIS had a higher prevalence of International Standard Classification of Education (ISCED) ≥6 (men: 67.2–69.3%, women: 43.6–44.2%) than those from the CSLC data (men: 36.7%, women: 21.1%).

Table S1 shows the numbers of male offenders and non-offenders according to their characteristics and the results of Pearson's $\chi^2$ test and multivariable logistic regression analysis. Among men, self-reported physical child abuse was significantly associated with "work from home 4–7 days/week (Odds Ratios [ORs] = 1.81, $P$ value [P] = 0.028)," "decrease of work (ORs = 1.66, $P$ = 0.025)," "normal relationships with household members (compared to good relationships, ORs = 1.57, $P$ = 0.039)," "COVID-19

infected, both themselves and household members, within a year (ORs = 4.10, $P = 0.007$),” “unwillingness to receive COVID-19 vaccination because the license process of the vaccine is doubtful (ORs = 2.85, $P = 0.006$),” “high levels of benevolent sexism (ORs = 1.65, $P = 0.018$),” “history of physical child abuse (ORs = 2.96, $P < 0.001$),” and “history of verbal child abuse (ORs = 2.24, $P = 0.004$).” All VIF values were less than 1.5, indicating a lack of collinearity among predictor variables.

Table S2 shows the number of female offenders and non-offenders according to their characteristics and the results of Pearson's $\chi^2$ test and multivariable logistic regression analysis. Among women, self-reported physical child abuse was significantly associated with “bad relationships with household members (compared to good relationships, ORs = 3.51, $P < 0.001$),” “strong fear of COVID-19 (ORs = 1.44, $P = 0.028$),” “COVID-19 infected, both themselves and household members, within a year (ORs = 3.22, $P = 0.038$),” “feelings of discrimination related to COVID-19 in the past two months (ORs = 2.08, $P = 0.044$),” and “history of verbal child abuse (ORs = 2.66, $P < 0.001$).” It was significantly and negatively associated with “aged 50-59 (compared to aged 20-29, ORs = 0.29, $P = 0.029$)” and “unwillingness to receive COVID-19 vaccination for other reasons than the license process of the vaccine is doubtful (ORs = 0.55, $P = 0.037$).” All VIF values were less than 2.2, indicating a lack of collinearity among predictor variables.

# DISCUSSION

## The importance of self-reported child abuse

Compared to officially reported child abuse data, self-reported data may include a greater number of milder cases. However, society is busy with specific responses to address previously reported severe cases. The number of officially reported child abuse cases is increasing across developed countries, while the number of cases identified or suggested as child abuse has not changed significantly (*Ministry of Health, Labour and Welfare in Japan, 2022*; *Department for Education, 2019*; *Children's Bureau, 2020*; *Australian Institute of Health and Welface, 2020*). This may indicate that people are more conscious of abuse or that mild cases are actually increasing, but society has a limited capacity to respond to all officially reported cases. Although mild violence is not necessarily consistent with severe violence, it is a risk factor for difficult cases in previously reported data.

## The validity of our sample

As Table 1 shows, the caregivers analyzed in our study cohort had a similar population distribution with regard to sex and age, compared with those from the CSLC data, while the non-analyzed caregivers did not. This may have been caused by a trend of younger people being more likely to respond "Unknown/Do not want to answer" to the question about physical child abuse. While we cannot determine how many offenders answered "Unknown/Do not want to answer," these options are factors that support the quality of submitted data. In this web-based survey, panelists had to fill in all the questions to submit the data, and these additional options prevented them from giving contradictory answers, and enabled us to set the exclusion criteria. In addition, web-based

surveys are reported to have better quality responses to delicate questions than do face-to-face interactions (*Hasegawa, 2007*). Our sample was biased toward people with higher educational backgrounds, similar to a previous report comparing the CSLC data and Internet surveys (*Taniguchi & Omori, 2022*). Considering the relationship between low educational background and child abuse, our results may be adaptable to a group in which child abuse is unlikely.

## Common risk factors for both genders

The common risk factors of self-reported physical child abuse in offenders of both genders are "COVID-19 infected, both themselves and household members, within a year," "poor relationships with household members," and "history of child abuse." The intergenerational cycle of child abuse is known worldwide as a major risk factor (*Berlin, Appleyard & Dodge, 2011*; *Horikawa et al., 2016*; *Assink et al., 2018*; *Anderson et al., 2018*). In this pandemic, "COVID-19 infected, both themselves and household members, within a year" had a high odds ratio, exceeding that of "history of child abuse." Considering that "only themselves being infected" did not show a high prevalence of self-reported physical child abuse, depression symptoms due to COVID-19 or Long COVID would not significantly change their responses. Therefore, physical child abuse is very likely to occur in a household where the offender and someone else were infected within a year. Considering the high odds ratios exceeding those of "history of child abuse," abuse is more likely to occur during periods of physical, mental, or social stress, when household members are frustrated with each other. For offenders, this stress may differ between "only offender infected (with COVID-19)" and "Offender and another household member infected." For example, it is difficult to determine the source of the infection in most cases. When several family members are simultaneously infected, they may often assume that others might be responsible for the infection. Some strong risk factors were found like intergenerational chain of child abuse and low socioeconomic status, but practical intervention is not possible. The negative impacts of COVID-19 have been widely studied, and this result might be one of the risks for increased stress, especially toward their families in particular situations (*Amin & Parveen, 2022*).

## Male offenders

Risk factors specific to male offenders were "work from home 4–7 days/week," "decreased work," "unwillingness to receive COVID-19 vaccination because the license process of the vaccine is doubtful," and "high benevolent sexism scale." Benevolent sexism measures the attitude toward stereotyped gender roles; men should work, and women should do household chores. We estimate that high levels of benevolent sexism of men reflect an emphasis on the importance of an occupation, leading to the strong stress caused by "work from home 4–7 days/week" and "decreased work." The strength of gender roles is inferred from the gender gap in the labor force participation. Substantial gaps in labor force participation prevalence between married men and women (25–30%) were observed in countries such as Japan, South Korea, Italy, and Brazil, compared to the gaps (less than 15%) in countries such as France, Sweden, Finland, England, and Denmark (*ILOSTAT,*

*2022*). In Japan, half of the employed women quit their jobs after childbirth (*Ministry of Health, Labour and Welfare in Japan , 2021*). As Japan has very established gender roles, it is unclear whether the work-related risk factors in our results are also found in countries with gender roles as prominent, such as in Italy and Brazil. Notably, low telework penetration is consistent with high gender gap in the labor force participation to some extent. It is reported that telework penetration before the pandemic was low in Japan, Italy, and Brazil and high in France, Sweden, Finland, England, and Denmark (*OECD, 2021*). The pandemic has increased telework globally, but it was a rapid change for the country with prominent stereotyped gender roles, such as Japan. Moreover, "work from home 4–7 days/week" and "decreased work" are associated with increased time spent at home and anxiety about potential job loss. For men with strong stereotyped gender roles, the sudden increase in time with their children is largely inconsistent with their previous lifestyles. This may be a primary cause of dissatisfaction with their children and spouses because they were previously uninvolved in childcare. In addition, the situation where the husband is the only working member can increase their dissatisfaction with their family when faced with the uncertainty of unemployment. It is reported that "working remotely" and "becoming unemployed during the COVID-19 pandemic" were not associated with anxiety symptoms in Finland (*Savolainen et al., 2021*). The telework system introduced during the pandemic will likely continue in many companies. Globally, as there was no precedent for the sudden increase in time at home for men, it is difficult to speculate how long the estimated risk factors due to men's familial dissatisfaction will continue in countries with strong gender roles.

Anxiety over decreasing or losing jobs is also influenced by financial support in each country. Many developed countries have job retention schemes and support the income of workers who are temporarily unemployed or whose work hours have been decreased. Japan, England, and Germany showed the lowest unemployment rates before and during the pandemic (*OECD, 2020*). On the other hand, in Japan, wages supported by job retention schemes for workers with 100% reductions in work hours are much higher than the amounts provided by unemployment benefits. In countries with large differences, such as in Japan, Portugal, Denmark, and Poland, anxiety surrounding job loss might be high (*OECD, 2020*).

"Unwillingness to receive COVID-19 vaccination because the license process of the vaccine is doubtful" was a risk factor for male offenders. Considering that "fear of COVID-19" and "feelings of discrimination related to COVID-19 infection" were risk factors associated with female offenders, men with no vaccination owing to skepticism might be subject to domestic discord due to their uncooperative attitude toward infection prevention.

## Female offenders

Risk factors specific to female offenders were "bad relationships with household members (compared to good relationships)," "strong fear of COVID-19," and "feelings of discrimination related to COVID-19 in the past two months." Unlike the discussion on the work-related risk factors in men, the infection-related risk factors are not always explained

by gender roles. In Northern European countries, women showed a higher prevalence of anxiety and depression symptoms due to the COVID-19 pandemic (*Savolainen et al., 2021*; *Johansson et al., 2013*). Importantly, infection-related factors are gradually reduced with the abatement of the pandemic.

This study had some limitations. First, we found that self-reported physical child abuse was likely to occur in the household where the offender and someone were infected within a year, with a high odds ratio similar to "history of child abuse." However, we could not identify who the someone is, and a particular combination might influence the relationship with child abuse. For example, while the combination of infected offender and child might indicate a simple mechanism of child abuse, the combination of infected offender and spouse might indicate a complex mechanism of child abuse regarding childcare, partly explained by the strength of gender roles. Second, we could not determine whether the offenders were biological parents or parents-in-law because there was no definition of parents in JACSIS. It has been reported that parents-in-law abuse children more than biological parents, and different types of abuse are observed (*Turner et al., 2013*; *Daly & Wilson, 1994*; *Baba et al., 2020*). Third, this was a cross-sectional study for which causality was not identified. Fourth, although the analyzed respondents were representative of the population in terms of age and sex, they may have biased the characteristics and influenced the results. Our sample included many people with high educational backgrounds, similar to a previous study (*Taniguchi & Omori, 2022*). Respondents may have had milder personalities than the average for the population, which may have in turn resulted in a mild and low prevalence of self-reported child abuse.

## CONCLUSION

Physical child abuse was very likely to be reported in the "COVID-19 infected, both themselves and household members, within a year" households. Among male offenders, a significant relationship was observed in the change in work situations, such as "work from home 4–7 days/week" and "decreased work." Difficulty adapting to work-related changes and the fear of losing jobs may be high in countries with established gender roles and insufficient financial support. Among female offenders, a significant relationship was observed regarding fear of infection, which is consistent with the findings of previous studies on other pandemics (*Savolainen et al., 2021*; *Johansson et al., 2013*). When emergencies, such as pandemics, keep people at home or change past lifestyles, the parental stress might be more prevalent in countries with strong stereotyped gender roles.

## ACKNOWLEDGEMENTS

We thank all members of the JACSIS Survey Team. We would like to thank Honyaku Center Inc. for English language editing.

### Funding

This study was supported by the Japan Society for the Promotion of Science KAKENHI Grants [grant number 21H04856, 20K13721, 19K10446, 18H03107], the Health Labour Sciences Research Grant [grant number 19FA1005; 19FA1012], grants from Chiba Foundation for Health Promotion & Disease Prevention, Innovative Research Program on Suicide Countermeasures (R3-2-2), the Ministry of Health, Labour and Welfare Special Research Program Grant [grant number JPMH20CA2046] and the Japan Agency for Medical Research and Development [AMED; grant number 2033648]. The funders had no role in study design, data collection and analysis, decision to publish, or preparation of the manuscript.

### Grant Disclosures

The following grant information was disclosed by the authors:
Japan Society for the Promotion of Science KAKENHI Grants: 21H04856, 20K13721, 19K10446, 18H03107.
Health Labour Sciences Research Grant: 19FA1005, 19FA1012.
Chiba Foundation for Health Promotion & Disease Prevention, Innovative Research Program on Suicide Countermeasures: (R3-2-2).
Ministry of Health, Labour and Welfare Special Research Program Grant: JPMH20CA2046.
Japan Agency for Medical Research and Development AMED: 2033648.

### Competing Interests

The authors declare there are no competing interests.

### Author Contributions

- Haruaki Naito and Katsuya Nitta analyzed the data, prepared figures and/or tables, authored or reviewed drafts of the article, and approved the final draft.
- Yasuhiro Kakiuchi conceived and designed the experiments, analyzed the data, authored or reviewed drafts of the article, and approved the final draft.

### Human Ethics

The following information was supplied relating to ethical approvals (i.e., approving body and any reference numbers):

The Research Ethics Committee of the Osaka International Cancer Institute granted Ethical approval.

### Data Availability

The raw data is available in the Supplementary File.

### Supplemental Information

Supplemental information for this article can be found online at http://dx.doi.org/10.7717/peerj.15346#supplemental-information.

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
