# Peer review of "Risk factors of self-reported physical child abuse during the COVID-19 pandemic in Japan: work-related changes in men and fear of COVID-19 in women"

_PeerJ, doi:10.7717/peerj.15346_

## Round 0.1 · original submission · Major Revisions

We have received reviews from the reviewers for your manuscript. This is a well-written article; however, some areas need improvements. Please include a paragraph in the Discussion section – with regard to the implications of the findings for clinical audiences.

·

Basic reporting

The article conveys a clear message on child abuse and its implication during Covid -19. The factors tested in the study are a value addition to the literature, especially in a pandemic situation where quarantine is practised. Overall there is a novelty in the approach and reporting of the study.

Experimental design

The study design is of any standard scientific design in health studies. However, all the participants included in the study need to be defined. The elaboration of inclusive criteria and exclusion criteria in an internet survey needs precision reporting. Hence, the duplication of this study in different conditions can be done.

Validity of the findings

The results and raw data reporting were correct and per standard. More pictorial depictions and table are needed for the presentation of the result.

Additional comments

The can discuss the matter considering the social conditions during covid 19. The current discussion gives some ideas about the impact of various factors on child abuse.

Reviewer 2 ·

Basic reporting

Overall, the article is written in clear and professional English. However, there are a number of small errors in grammar which should be corrected (Lines 24, 64, 67-68, 126, 141, 219, 267, 271, and 351).

Limited references were provided, but this largely a function of the emerging research in the area of pandemic-related child abuse. The background provided little information regarding the validity of self-reported child abuse and it is recommended that this be addressed in greater depth, potentially including it in the discussion section to compare to the present study's findings.

While the article is appropriately structured and all relevant files included for review, the values of the associations identified should be reported in the text, not only the tables.

Experimental design

The study design appears acceptable, but there is little information included in the methods to clarify how sampling of the panelists was conducted. The first paragraph both describes the selection as representative of the national population, but also as randomly sampled, so there is clearly a need for greater detail. Furthermore, there should be description of how the questionnaires were delivered and whether this occurred all at once or staggered throughout the enrollment period or perhaps in waves in response to low participation. Providing the number of caregivers approached in the text would be useful, as would a later discussion regarding the potential reasons for lack of participation and whether there was any identifiable difference between eligible respondents and non-respondents. Were both groups representative of the national population of caregivers?

Additional recommendations regarding the methods section include clarifying whether the questions were open-ended or had pre-defined response categories, providing a more detailed description or definition of benevolent sexism, and improving the section on the statistical analyses conducted.

With regard to the analyses, there are a number of questions. Were any interaction terms considered or assessed? Was there any assessment of collinearity among variables? (Some of the variables seem likely to correlate closely with others.) It appears that the chi-squared analysis was conducted as a test of the overall variables, then the univariable logistic regression conducted to identify the specific response levels demonstrating association. The results do not need to be presented on separate tables, but could easily be combined in order to improve reporting clarity. Furthermore, given the number of significant variables identified, why was no multivariable model considered? This would provide a more complete picture of the relative impact of each variable in this complex context.

As mentioned previously, the results should include brief reporting of the associations within the text. Finally, the results would benefit from additional clarification regarding the reporting of overall variable significance versus specific response level significance since this is not specified in the methods and the similar terminology between paragraphs on Tables S1 and S2 requires careful re-reading and examination of the tables to understand thoroughly.

Validity of the findings

The discussion is of appropriate length, but would benefit from re-structuring to improve readability. Since male and female respondents were modeled separately, it is appropriate to report and discuss each separately, yet despite the sub-headings the discussion is not clearly delineated, moving back and forth. Additionally, the limitations section did not address the potential problems with panel surveys or self-selection of sampled panelists for participation.

The conclusions appear to step beyond the "reach" of the data, ascribing causality (line 25, "lead to"; line 53, "due to") where none has been assessed. There certainly appear to be meaningful conditions which impact the likelihood self-reported child abuse. However, the unclear representativeness of the survey respondents with regard to the national population of caregivers makes it difficult to accept the findings as predictive or useful for directing public health policy. It would also be beneficial to broaden the discussion (and hence the conclusions drawn) to consider variation by country in financial support which may have significantly influenced the levels of anxiety associated with job losses, rather than just the established gender roles.

Additional comments

The importance of the topic is undeniable and the authors are to be commended for the use of an existing survey panel to address the question. Further refinement of the text and potentially the creation of multivariable models are recommended.

·

Basic reporting

clear and professional English.



I think literature are not that sufficient, also references.



Flowchart of the analyzed participants selection ok.



No comment.

Experimental design

Yes with in scope of Journal


No comment


No comment



No comment

Validity of the findings

No comment



Data Available





I think Conclusions needs correction, author has not mentioned any hypothesis but in conclusion he is writing a hypothesis developed?

Additional comments

I think you need to add some more references.
you can refer to, "Amin UA, Parveen AP. Impact of COVID-19 on children. Middle East Current Psychiatry, Ain Shams University. 2022;29(1):94. doi:10.1186/s43045-022-00256-3"
Line no 300, reference; add name of Journal and date when assessed.

·

Basic reporting

This study examines the Risk factors of self-reported physical child abuse
during the COVID-19 pandemic in Japan. The fact that this is one of few studies from this area makes it an important study.

Introduction – Line 73-74
"To the best of our knowledge, no study in Japan has identified a significant relationship between physical child abuse and pandemic-related factors of offenders."

Its best to place this line at the begining of the discussion or at the end of the introduction section.


Methods - Line 89 - 93
"The survey focused on public health issues regarding COVID-19 pandemic and was conducted by an Internet research company (Rakuten Insight, Inc.), which had 2.3 million registered panelists. From September 27 to October 29, 2021, questionnaires were distributed to 59,219 panelists to represent the Japanese population regarding age, gender, and residential prefecture using a simple random sampling process (the Japan COVID-19 and Society Internet Survey, 2022)."

It is better to use words that represent the aim of the sentence of the author.


Results - Line 145-148
"Among the caregivers with child aged 0-14 years, a total of 134 men answered Yes and 2,034 answered 'No' to the question of physical child abuse in the past two months. A total of 187 women answered 'Yes' and 2,038 answered No."

Child aged 0-14 is a little weird. Please correct it with using less than one year.

Experimental design

This study is within the scope of the journal. The research questions is clearly and relevantly defined. Research gap is clearly stated.
The authors used an already existed dataset which is publicly available that has already took care of the participants identity. Study method is described properly.

Validity of the findings

The tables are presented clearly and reporting is meaningful.
Statistical analysis used to report the result of this study is clearly stated.

---

## Round 0.2 · Minor Revisions

Dear Authors,

I am pleased to inform you that your paper entitled "Risk factors of self-reported physical child abuse during the COVID-19 pandemic in Japan: Work-related changes in men and fear of COVID-19 in women" has received reviews from four reviewers. Please address the comments of Reviewer #2.

·

Basic reporting

The report has been improved by incorporating the comments.

Experimental design

Fine and explained according to the requirements.

Validity of the findings

Fine and explained according to the requirements.

Additional comments

fine

Reviewer 2 ·

Basic reporting

The authors have immensely improved the manuscript. The literature cited is appropriate and the content is much clearer than before.

However, there are a few remaining errors in English grammar that should be corrected prior to publication:
Line 27 - "which factors is strongly associate" --> subject-verb agreement
Line 32 - "the Internet survey from September..." --> needs a verb, such as "conducted"
Line 33 (and repeated uses throughout manuscript) - "aged less than 1 year to 14 years" --> substitute simply "aged less than 14 years"
Line 37 (and repeated uses throughout manuscript) - "univariate" and "multivariate" --> since there is only a single outcome variable in each model, the correct terms would be "univariable" and "multivariable" (for a detailed explanation of the difference, please refer to Hidalgo and Goodman in AJPH, https://ajph.aphapublications.org/doi/abs/10.2105/AJPH.2012.300897)
Line 183 - spelling error "mele" instead of male
Line 279 - "law telework" --> should be "low"
Table 1 - "Woman" --> should be "Women"

Please provide the definition of JASTIS as was provide for JACSIS.
Additionally, in the methods section, ensure that Figure 1 is referenced and please provide the final total number of participants in Line 114.

Experimental design

The research design is much clearer in the updated manuscript, as are the methods, particularly the details of the participant selection. One recommendation is that Line 202 be expanded to describe the use of the VIF, for example, "All VIF values were less than 1.5, indicating a lack of collinearity among predictor variables." Additionally, the use ofo VIF should be clarified in the methods.

Validity of the findings

No comment

·

Basic reporting

All comments were sufficiently addressed in the current revision.

Experimental design

All comments were sufficiently addressed in the current revision.

Validity of the findings

All comments were sufficiently addressed in the current revision.

---

## Round 0.3 · accepted · Accept

The authors have addressed all of the reviewers' comments. The manuscript may be accepted as per the Journal Policy.

Reviewer 2 ·

Basic reporting

All areas of concern have been addressed. My only remaining wonder is why the authors have chosen to keep the main results tables as supplemental rather than part of the main paper. Recommend including them as Tables 2 & 3 rather than S1 & S2 in order to ensure the article is self-contained.

Experimental design

No comment

Validity of the findings

No comment